# Chemosensory Proteins (CSPs) in the Cotton Bollworm *Helicoverpa armigera*

**DOI:** 10.3390/insects13010029

**Published:** 2021-12-27

**Authors:** Aniruddha Agnihotri, Naiyong Liu, Wei Xu

**Affiliations:** 1Food Futures Institute, Murdoch University, Murdoch, WA 6150, Australia; ar.agnihotri@murdoch.edu.au; 2Key Laboratory of Forest Disaster Warning and Control of Yunnan Province, Southwest Forestry University, Kunming 650224, China; naiyong_2013@163.com

**Keywords:** chemosensory protein, CSP, cotton bollworm, *Helicoverpa armigera*, protein expression, olfactory

## Abstract

**Simple Summary:**

The insect chemosensory system is crucial in regulating insect behaviors. Chemosensory proteins (CSPs) are a family of small, soluble proteins conventionally known to transport odorant molecules in insect chemosensory system. Besides chemosensation, CSPs have been reported to play important roles in development, nutrient metabolism, and insecticide resistance. Therefore, identification and characterization of previously unknown CSPs will be valuable for further investigation of this protein family. The cotton bollworm, *Helicoverpa armigera* (Hübner) is among the most serious insect pests in various agricultural and horticultural crops. In this study, 27 CSP genes were identified from *H. armigera* genome and transcriptome sequences, and their expression patterns were further examined by using transcriptomic data obtained from different tissues and stages. The results demonstrate that *H. armigera* CSP genes are highly expressed in both chemosensory and non-chemosensory tissues. Moreover, a new recombinant expression method was developed that can significantly increase *H. armigera* CSP expression levels as soluble proteins in *Escherichia coli*. This study improves our understanding of insect CSPs and developed a new approach to highly express recombinant CSPs, which can be expanded to examine CSPs in other species for functional characterization.

**Abstract:**

Chemosensory proteins (CSPs) are a family of small, soluble proteins that play a crucial role in transporting odorant and pheromone molecules in the insect chemosensory system. Recent studies reveal that they also function in development, nutrient metabolism and insecticide resistance. In-depth and systematic characterization of previously unknown CSPs will be valuable to investigate more detailed functionalities of this protein family. Here, we identified 27 CSP genes from the genome and transcriptome sequences of cotton bollworm, *Helicoverpa armigera* (Hübner). The expression patterns of these genes were studied by using transcriptomic data obtained from different tissues and stages. The results demonstrate that *H. armigera* CSP genes are not only highly expressed in chemosensory tissues, such as antennae, mouthparts, and tarsi, but also in the salivary glands, cuticle epidermis, and hind gut. HarmCSP6 and 22 were selected as candidate CSPs for expression in *Escherichia coli* and purification. A new method was developed that significantly increased the HarmCSP6 and 22 expression levels as soluble recombinant proteins for purification. This study advances our understanding of insect CSPs and provides a new approach to highly express recombinant CSPs in *E. coli*.

## 1. Introduction

A myriad of insect behaviors such as foraging, host-feeding, mating and oviposition [1] are mediated by their chemosensory system. Various chemosensory sensilla distributed over the surface of chemosensory tissues are utilized to detect chemical signals from the environment [2,3,4]. When hydrophobic semiochemicals reach the aqueous sensillum lymph through the pores on chemosensory sensilla, it is difficult for them to reach the membrane-bound receptors directly due to their lack of solubility. Two classes of proteins are highly expressed in chemosensory tissues, where they help deliver the semiochemicals to the receptors for sensation [5], which are odorant-binding proteins (OBPs) [6] and chemosensory proteins (CSPs) [7]. OBPs are one family of small (14–20 kDa), soluble proteins with six conserved cysteines and are involved in odorant binding and delivery in the sensilla [7,8]. CSPs do not share sequence similarities with OBPs and are characterized by only four conserved cysteines [9,10,11,12]. The first CSP protein was found to increase significantly in regenerating legs of the American cockroach, *Periplaneta americana*, and decrease significantly when leg regeneration was complete [13]. OS-D, a similar CSP, was cloned later from *Drosophila melanogaster* antennae and assumed to play a role in pheromone binding [14]. The completion of the various insect genome sequences has led to the recent identification and characterization of more insect CSP genes [7,15]. The numbers of CSPs vary in different insect species. For example, only 4 CSPs were identified in *D. melanogaster*, 6 in *Apis mellifera* [16], and 20 in *Bombyx mori* [17], whereas 70 were found in *Locusta migratoria* [18].

Many CSP members have been abundantly detected in the chemosensory sensillum lymph and display binding activity for various odorants [11]. For example, one CSP identified from the antennae of both male and female *Helicoverpa armigera* adults (namely HarmCSP6) exhibits high binding affinities for pheromone components [19]. This suggests that HarmCSP6 may function as a pheromone-binding protein and be involved in delivering sex pheromone components [19]. A CSP was detected in *Cactoblastis cactorum* maxillary palps and may play a role in sensing carbon dioxide [20]. Not all CSPs are expressed in chemoreception organs. For example, a recent proteomic study on *H. armigera* eyes identified a few CSPs, which may deliver compounds required for vision across aqueous biological fluids, such as the carotenoids and their breakdown products [21]. Another study showed that a CSP (namely HarmCSP4) is highly and exclusively expressed in the proboscis of *H. armigera* or *Helicoverpa assulta*. When moths sucked on a sugar solution, it was found that HarmCSP4 is partly extruded from the adult proboscis and may act as a wetting agent to decrease the surface tension of aqueous solutions and, thereby, the pressure involved in sucking [22].

Various insect CSPs have been successfully expressed and purified in the *Escherichia coli* expression system, which grants us an opportunity to utilize CSPs to develop a reverse chemical ecology approach for screening new candidate attractant or repellent chemicals [23]. It is a new approach for identifying target ligands based on the binding ability of CSPs or OBPs to test compounds, which may be efficient in the identification of a more manageable set of candidate compounds. However, this approach requires a large amount of purified recombinant CSPs, so the high expression and fast method to produce recombinant insect CSPs is extremely helpful for the reverse chemical ecology approach to screen candidate ligands.

*Helicoverpa armigera* is among the most polyphagous and cosmopolitan insect pests and is considered one of the most destructive agricultural pest species. It feeds on hundreds of plants, including various agricultural crops such as cotton, peanuts, soybeans, maize, and tomatoes. Previous studies from different labs altogether identified 21 CSPs from *H. armigera* [19,24,25,26,27]. The publication of the *H. armigera* genome sequences supplies an invaluable resource [28] for us to identify new CSPs and study their specific features [28].

In this study, we aimed to (1) identify and characterize the CSPs from *H. armigera* and (2) develop a more efficient and productive method to express CSPs. The first aim will assist in linking the potential functions or candidate ligands to identified CSPs. The second aim will help provide plenty of purified recombinant CSPs for investigating CSP–ligand interactions. Both aims will improve our understanding of insect CSPs and may help discover new candidate attractants or repellents through a reverse chemical ecology approach [23]. The results showed that a total of 27 CSP genes were annotated from *H. armigera*, including 6 new CSPs. Their expression patterns were further analyzed by using transcriptome data obtained from different development stages and tissues. Moreover, a new method was developed to express recombinant HarmCSPs from the *E. coli* system, increasing the expression in soluble form and facilitating the purification, which can be applied to studies of other insect CSPs.

## 2. Materials and Methods

### 2.1. Insect Materials

*Helicoverpa armigera* pupae were provided by University of Queensland [29] in August, 2016 and reared at 26 ± 2 °C and 70% humidity [30]. Twenty male or female antennae were collected from adults aged 1–3 days after they emerged and kept in RNAlater (Invitrogen, Waltham, MA, USA) immediately for extracting total RNA samples by using the Qiagen RNeasy mini kit (Qiagen, Valencia, CA, USA). The extracted total RNA was treated using DNaseI (New England Biolabs, Ipswich, MA, USA) for 30 min at 37 °C. Then, the treated total RNA samples were quantified and quality controlled using a NanoDrop ND-2000 (Thermo Scientific, Waltham, MA, USA). These RNA samples were then used for cloning *H. armigera* CSP genes of interest for expression study.

### 2.2. Bioinformatics and Phylogenetics Analysis

Genes encoding CSPs in previously published *H. armigera* genome assemblies [28] were identified using TBLASTN searches with reported *B. mori*, *H. armigera,* and *H. assulta* CSPs using a query with an E-value cutoff of 1e^−^^5^. Extensive manual curation was then performed on the *H. armigera* WebApollo (http://webapollo.bioinformatics.csiro.au/helicoverpa_armigera) accessed on 10 October 2021 by using transcriptomic data as well as GT–AG rule for exon/intron splice sites in 2018. The annotated *H. armigera* CSP amino acid sequences were validated using NCBI blast to obtain the identities and similarities to orthologous genes from other species. *H. armigera* CSP amino acid sequences identified in this study are available in Appendix A, which were further used to find CSP genes in the *Helicoverpa zea* genome [28] and the *Helicoverpa assulta* transcriptome [31] data. Encoded proteins were aligned (Appendix A) using default settings in Geneious 8.0.5 (http://www.geneious.com) accessed on 10 October 2021 [32]. Signal peptides at the N-terminus of the identified HarmCSPs were predicted online using SignalP 4.1 (http://www.cbs.dtu.dk/services/SignalP) accessed on 10 October 2021. The molecular weight (MW) and isoelectric point (pI) of each mature HarmCSP were calculated using ExPASy Protoparam server (http://www.expasy.org/tools/protparam.html) accessed on 10 October 2021. HarmCSP protein sequences were utilized searching the top blast hit sequences by using blastp from NCBI (Appendix A). Graphic Marker (http://www.wormweb.org/exonintron) was used accessed on 10 October 2021 to generate the Exon–Intron graphics of HarmCSPs.

Selected lepidopteran CSP amino acid sequences (*B. mori*, *H. armigera*, *H. assulta*, *H. zea*, *P. xylostella*, *D. plexippus*, *H. melpomene*, *Manduca sexta*, and *Heliothis virescens*) were utilized to produce an entry file for the phylogenetic tree in Geneious alignment (Geneious software 8.05) with gap opening penalty (10.00) and gap extension penalty (0.10) for multiple sequence clustal alignment (30% delay divergent cutoff). Then a maximum-likelihood tree was built based on Jones–Taylor–Thornton model with partial deletion and site coverage cutoff (70%) using the default settings (Figure 1).

### 2.3. Expression Profiles of HarmCSPs

In the previous study [28], total RNA samples from the following *H. armigera* tissues were sequenced and deposited into NCBI, including mouthparts (6,608,663), antennae (GenBank BioSample 6,608,662), fat body (6,608,648), epidermis (6,608,657), midgut (6,608,643), foregut (6,608,642), hindgut (6,608,644), hemocytes (6,608,650), Malpighian tubules (6,608,645), trachea (6,608,653), hearts (6,608,652), silk glands (6,608,646), salivary glands (6,608,651), ventral nerve (6,608,649), and muscle (6,608,654) of the 5th instar larvae; female antennae (6,608,659), male antennae (6,608,658), female heads (with antennae) (6,608,640), male heads (with antennae) (6,608,641), female tarsi (6,608,645), male tarsi (6,608,664), female thorax (6,608,661), male thorax (6,608,660), female abdomens (with ovaries) (6,608,639), male abdomens (with testes) (6,608,638), female ovaries (6,608,655) and male testes (6,608,656) from day 0 to day 5 adults; embryo (6,608,634), 3rd instar larvae (6,608,635), post feeding larvae (6,608,636) and whole pupae (6,608,637). Finally, in silico expression profiles of HarmCSPs were prepared using DEW (http://dew.sourceforge.net/, accessed on 10 October 2021) as described before [31].

### 2.4. Gene Cloning

All *H. armigera* cDNA templates were generated from extracted total RNA samples using the SuperScript^®^ VILO™ cDNA Synthesis Kit (Invitrogen, Waltham, MA, USA), following the manufacturer’s instruction. Gene-specific primers for the full-length HarmCSP6 and 22 ORF sequences, including HarmCSP6-F, ATGAAATTCGTGTTAGTACTG; HarmCSP6-R, TTATTCGGGGATCTGGATGCCGTTG; HarmCSP21/22-F, ATGAACTCTGCTATCGTGCTATG and HarmCSP21/22-R, TTAATGTTGCACTTCTTCGAGCTCC, were designed and ordered from Macrogen (Seoul, South Korea). Phusion High-Fidelity DNA Polymerase (New England Biolabs, Ipswich, MA, USA) was used for the PCR amplification. The PCR program was 95 °C for 3 min; followed by 30 cycles at 95 °C for 30 s, 55 °C for 30 s, and 72 °C for 30 s as well; and a final extension at 72 °C for 10 min. QIAquick gel extraction reagents (Qiagen, Valencia, CA, USA) were used to purify amplified PCR products, which were further ligated to the pBluescript SKII (–) vector (Stratagene, La Jolla, CA, USA) using blunt end ligation at the EcoRV site of the vector. The ligation products were transformed into competent *Escherichia coli* DH5α cells (New England BioLabs, Ipswich, MA, USA). Three randomly picked colonies were selected for plasmid purification and DNA sequencing to confirm the correctly constructed vectors with candidate HarmCSP genes.

The constructed pBluescript SKII (–) vectors containing the full-length ORF of HarmCSP6 and HarmCSP22 were isolated and the restriction digestion, NcoI (forward) and SacI (reverse), was performed to obtain the complete gene with desired restriction cites for ligation in pET30a (+). Post ligation, the recombinant pET30a (+)-HarmCSP6 and pET30a (+)-HarmCSP22 vectors were then transformed using the heat shock method into a One Shot^TM^ Top 10 *E. coli*–competent cell strain (Invitrogen, Waltham, MA, USA). The positive colonies containing the recombinant vector were selected based on colony PCR using vector-specific T7 primers. The selected colonies were further inoculated in 5 mL LB (with kanamycin) over night at 37 °C for plasmid isolation. Plasmids were isolated from these cultures using a Qiagen Miniprep Plasmid Isolation Kit (Qiagen, Valencia, CA, USA). Isolated plasmids were further sent for sequencing to confirm the sequence of HarmCSP6 and HarmCSP22 genes ligated in them.

### 2.5. In Vitro Expression Study of HarmCSP6 and HarmCSP22 in a Bacterial System

Recombinant pET30a(+) plasmids containing the correct HarmCSP6 and HarmCSP22 genes were further transformed into One Shot^TM^ BL21-DE3 *E. coli*–competent cells (Invitrogen, Waltham, MA, USA) for the in vitro expression of HarmCSP6 and HarmCSP22. Positive colonies of transformed BL21-DE3 *E. coli* cells containing pET30a(+)-HarmCSP6 and pET30a(+)-HarmCSP22 plasmids, respectively, were inoculated in 5 mL LB (with kanamycin) media and incubated over night for 16 h at 37 °C at 200 RPM to obtain the starter culture. Then, the next day, this starter culture was used to inoculate 5 mL M16 media in a 25 mL flask, with 0% (control), 0.5%, and 1.0% glycerol each. M16 media is a customized media developed in our lab containing the following components: 3.2% tryptone, 1.8% yeast extract, 0.25% NaCl, and 10% (*v*/*v*) phosphate buffer (0.72 M K_2_HPO_4_ + 0.17 M KH_2_PO_4_). Each media combination was inoculated in triplicate. Post inoculation, cultures were left on a bench top shaker for overnight (16 h) incubation at 37 °C and 200 RPM. Post 16 h, pre-induction OD600 and pH values of the culture were measured, and all cultures were induced for protein expression using a 250 µM final concentration of isopropyl β-d-1-thiogalactopyranoside (IPTG). Post induction, cultures were again incubated for 5 h at 37 °C and 200 RPM. Post incubation, final OD600 and pH values of each were measured. Further, *E. coli* cells were harvested using centrifugation (5000× *g*) at 4 °C for 10 min. Cell pellet was separated from the media and washed using 20 mM Tris buffer at pH 7.4 to remove the remnant media, which was then resuspended in a lysis buffer (20 mM Tris pH 7.4, 500 mM NaCl, and 1 mM EDTA). The resuspended cell pellet in lysis buffer solution was then sonicated for 5 min using s probe sonicator and then was centrifuged (30,000 RPM) for 45 min at 4 °C. Supernatant containing soluble cell extract was separated from the insoluble pellet and was further used for preparing samples for SDS PAGE (15%) analysis.

### 2.6. HarmCSP6 and HarmCSP22 Isolation, Purification, and Characterization

Large-scale recombinant expression of HarmCSP6 and HarmCSP22 was carried out using the same method as mentioned above. Each protein had one 6×His tag attached to its N-terminus and showed soluble expression inside the bacterial cell. Thus, the supernatant obtained from post cell lysis was directly used to purify by Ni-NTA HisTrap Gravity Affinity columns (GE Healthcare, North Richland Hills, TX, USA) as per the manufacturer’s protocol. Post purification, SDS PAGE was done to confirm the amount and purity of the eluted protein. The eluted proteins were then buffer exchanged and digested using enterokinase enzyme (New England Biolabs, Ipswich, MA, USA) to eliminate the N-terminus 6×His tag. Post digestion, proteins were again purified using Ni-NTA HisTrap Gravity Affinity columns (GE Healthcare, North Richland Hills, TX, USA) to separate the 6×His tag from the cargo protein. Further, for the structural characterization and to study the purity of digested and purified HarmCSP6 and HarmCSP22, small fractions of it were loaded on a C8 RP-HPLC column (Vydac^®^ 208TP54, C8, 5 μm, 4.6 × 250 mm) and HPLC was performed with a 1 mL/min flow rate and 10–50% ACN with a 0.1% aqueous TFA gradient over 25 min. The molecular weights of the final purified HarmCSP6 and HarmCSP22 proteins were confirmed using MALDI-TOF intact mass calculation using AutoFlex maX MALDI-TOF system (Bruker, Billerica, MA, USA). Furthermore, size-exclusion fast protein liquid chromatography (FPLC) analysis of both proteins was carried out using a Superdex 200 Increase 10/300 column (GE Healthcare, North Richland Hills, TX, USA) connected to a Bio-Rad biologic FPLC system (Bio-Rad, Hercules, CA, USA), with an isocratic flow of 0.75 mL/min buffer A (25 mM Tris-HCl pH 7.5 and 250 mM NaCl). Post characterization, both the proteins were stored in phosphate-buffered saline, pH 7.4, with 20% glycerol, 1 mM β-mercaptoethanol, and 1 mM EDTA at −80 °C until further use.

## 3. Results

### 3.1. Bioinformatics and Phylogenetics

A total of 27 HarmCSPs were identified from the *H. armigera* genome project [28]. Previously reported HarmCSP28 and 29 [28] are excluded from this study because neither of them contain the four conserved cysteines, the standard markers for insect CSPs. In this study, HarmCSP4, 8, 18, 19, 21, and 22 were firstly reported. HarmCSP21 and 22 are identical at both mRNA and amino acid sequence levels, although they are at two different genomic locations. In this work, we utilized a number of CSPs identified from previously reported insect genome or transcriptome projects including *B. mori* (20 CSPs) [17], *Manduca sexta* (21 CSPs) [33], *Danaus plexippus* (34 CSPs), and *Heliconius melpomene* (34 CSPs). We further constructed a phylogenetic tree only using lepidopteran CSPs from *B. mori* (20 CSPs), *P. xylostella* (15 CSPs), *M. sexta* (21 CSPs), *D. plexippus* (34 CSPs), *H. melpomene* (34 CSPs), *H. armigera* (27 CSPs), *H. zea* (27 CSPs), *H. assulta* (22 CSPs), and *H. virescens* (29 CSPs) (Figure 1). A *Helicoverpa/Heliothis*-specific CSP group and a butterfly-specific CSP group were detected. The *Helicoverpa/Heliothis*-specific CSP group includes HarmCSP21-25, HzeaCSP21-25, HassCSP23, HvirCSP21-25, and HvirCSP28, suggesting they may play specific roles in *Helicoverpa* species and *H. virescens* (Figure 1).

The alignment of HarmCSP, HzeaCSP, and HassCSP sequences (Appendix A) highlights the four conserved cysteine residues. Except the partial-length CSPs, most of the *Helicoverpa* CSPs share the distinctive hallmarks of the classic CSPs including an N-terminal signal peptide, small size (<160 amino acids), and four highly conserved cysteine residues known as “classic motif”. HarmCSP7 and HarmCSP16 are localized on scaffold 20, while the other 25 HarmCSP genes aggregate in a ~500,000-bp area on scaffold 119 (Figure 2a), implying that these HarmCSP genes likely evolved from the same ancestral gene. HarmCSP4, 7, and 16 contain three exons and two introns as shown by HarmCSP4 (Figure 2b). The other 24 HarmCSPs each contain two exons and one intron as shown by HarmCSP8 (Figure 2b).

The number of HarmCSP amino acid residues ranges from 107 (HarmCSP7 and 16) to 154 (HarmCSP8). A signal peptide within 16 to 19 amino acids was predicted at the N-terminus of all HarmCSPs except HarmCSP4 and HarmCSP19 (Appendix A). The molecular weights of predicted mature CSPs (without signal peptides) are from 10,092 to 16,417 Da. The isoelectric points (pI) of HarmCSPs were predicted to fall into either of two groups, acidic (<7.0, 17 HarmCSPs) and basic (>7.0, 10 HarmCSPs) (Appendix A).

### 3.2. Expression Profiles of H. armigera CSPs

The expression profiles of 27 HarmCSPs were examined based on the RNA-seq data of 31 different tissues ranging from embryo to adult stages. HarmCSP21 and 22 share the same RNA and protein sequences, so we used HarmCSP21/22 to represent their cumulative expression levels. In the expression heat map (Figure 3), white represents the low expression level (FPKM < 30) and yellow represents the middle expression level (FPKM = 200), while dark red represents the high expression level (FPKM ≥ 10,000).

Most *H. armigera* CSP genes were detected in multiple libraries (Figure 3). For example, HarmCSP1 was detected from the pupae, larval antennae, mouthparts, salivary glands, adult antennae, heads, thorax, tarsi, and male testes. HarmCSP21/22 was detected from the 3rd instar larva, post-feeding larva, larval foregut, hemocytes, salivary gland, heart, epidermis cuticle, antennae, mouthparts, adult tarsi, heads, antennae, and thorax. On the other hand, HarmCSP4 was only detected from adult antennae and heads (with antennae attached). HarmCSP7 was only detected from embryos and larval hearts. HarmCSP16 was specifically detected from the larval heart and trachea. HarmCSP18 and 19 both were only detected from larval antennae and mouthparts. Interestingly, HarmCSP8 was not detected in any tissue except for a low expression in the larval trachea.

Larval antennae and mouthparts are the major chemosensory tissues at the larvae stage, so they are supposed to be rich in CSPs. As expected, from these two tissues were found to have the highest numbers of CSPs in all larval tissues, including HarmCSP1, 2, 3, 5, 6, 9, 10, 12, 13, and 18–26. Similarly, in the larval epidermis/cuticle, HarmCSP9, 10, 12, 13, and 21/22–25 were detected. However, in the larval ventral nerve system, fat body, and Malpighian tubes only HarmCSP12 was detected. Importantly, a few CSPs showed extraordinarily high expression in salivary glands (HarmCSP21/22, 23, and 26), larval trachea (HarmCSP12), larval epidermis/cuticle (HarmCSP9, 13, and 21/22), larval antennae (HarmCSP13 and 21/22), and larval mouthparts (HarmCSP6 and 21/22).

We also studied the expression profiles of HarmCSPs in adult tissues for chemosensation (antennae and tarsi), reproduction (testes and ovaries), and in all three segments (heads, thoraxes, and abdomens). In general, more CSPs were detected in adult tissues than in larval tissues (Figure 3). HarmCSP1, 2, 4, 6, 9, 14, 15, 20, 21/22, 23, 24, and 26 were detected in both male and female antennae, suggesting they are the candidate carriers for the odorant delivery. Especially, HarmCSP4 was more highly detected from male antennae compared to female antennae, indicating it may be involved in sex pheromone binding. HarmCSP1, 2, 6, 21/22, and 26 are expressed much higher in adult heads (with antennae) than in adult antennae, indicating these HarmCSPs are possibly expressed in other tissues on the head, such as the eyes, palps, or proboscis. Surprisingly, many CSPs (HarmCSP1, 2, 3, 5, 6, 9, 12, 14, 15, 20, 21/22, 23, 24, 26, and 27) were highly detected in adult tarsi, the major gustatory organs [34]. This result suggests that CSPs may play a critical role in the insect taste system, for example, to deliver the tastant compounds from the host plants to the taste receptors. In male testes, HarmCSP1, 14, 15, and 26 were detected, while only HarmCSP11 was detected in female ovaries. 

### 3.3. In Vitro Expression Study of HarmCSP6 and HarmCSP22

We successfully cloned HarmCSP6 and 22 in pET30a (+) expression vector and transformed them into BL21-DE3 *E. coli* cells. HarmCSP6 was selected because it was highly expressed in adult tarsi, a major gustatory organ to contact the plant surface to help insects evaluate plants as food and oviposition sites. It was also detected at high expression levels in the larval antennae and mouthparts, suggesting that HarmCSP6 may be involved in insect–plant interactions. HarmCSP22 was highly expressed in larval salivary glands, mouthparts, cuticles, and antennae, so it may also play a role in perceiving plant compounds for chemosensation. Here, we developed a novel technique for the soluble expression of HarmCSP6 and HarmCSP22 in bacterial cells. M16 is a complex media mainly comprising enriched peptides (3.2% tryptone) as a prime carbon source for growth. M16 also contains, 1.8% yeast extract, which provides essential micronutrients for the growth of bacterial cells. It also contains phosphate buffer as buffering agent for pH stability and minimal salt (0.25%). For HarmCSP6 and 22 expression, M16 provided enriched nutrient to *E. coli* cells, resulting in very high bacterial growth and stable pH (Figure 4a,b). Further, the effect of glycerol on the protein expression of HarmCSP6 and HarmCSP22 was investigated. As seen in Figure 4, in presence of 0.5% glycerol, more than 90% of the total CSP6 and 22 expression was in their soluble form as compared to the control (no glycerol). Whereas, interestingly, in presence of 1% glycerol, for HarmCSP6 more than 95% of the expression was in the form of inclusion bodies and, for HarmCSP22, the total expression was less as compared to that of the control (Figure 4a,b). This indicated that 0.5% was the critical concentration of glycerol that not only enhanced the total protein expression but also converted most of the expression (>90%) from insoluble (inclusion bodies) to the soluble form.

Further, the addition of glycerol also helped the overall growth of bacterial cells and resulted in a higher cell density as compared to the control. The optical density at 600 nm for bacterial culture expressing HarmCSP6 and 22, grown in M16 media without glycerol, was 9.8 and 9.6, respectively. Whereas, in presence of 0.5% glycerol, this changed to 22.8 and 20.6, respectively (Figure 4c). Interestingly, in presence of 1% glycerol, a drop in the cell density to 19.4 and 15 was observed for HarmCSP6 and 22, respectively (Figure 4c). This drop correlated with the drop in protein expression, and thus, it can be concluded that, in presence of 1% glycerol, the bacterial growth as well as protein expression were downregulated, in both HarmCSP6 and 22. Moreover, this was also simultaneously observed with a change in pH of the culture in the presence of 1% glycerol. The pH was observed to be more acidic (≤6) as compared to that of the culture grown in control media or 0.5% glycerol-containing media. 

After purification, both HarmCSP6 and HarmCSP22 were observed to be >95% pure, based on SDS-PAGE gel as well as RP-HPLC results (Figure 5a,b,g). In addition, to confirm the structural integrity of expressed CSPs, final purified proteins were analyzed using size-exclusion FPLC. As seen in Figure 5c,d, both HarmCSP6 and 22 were expressed and purified as monomers (shown by black arrow), and no signs of aggregation were observed. 

MALDI-TOF mass spectrometry results revealed that both HarmCSP6 and HarmCSP22 were expressed and purified in intact form with no loss due to degradation or non-specific cleavages (Figure 5e,g). The difference between the theoretical intact mass and the observed intact mass of HarmCSP6 and 22 was well below the acceptable limit (error < 250 ppm), i.e., 2.1 and 2.39, respectively (Figure 5h). Thus, it can be concluded that the soluble expression and affinity purification of HarmCSP6 and 22 carried out in this study resulted in highly pure, stable, monomer and intact CSPs with no signs of degradation and aggregation.

## 4. Discussion

Since the first identification of insect CSPs in the 1990s, there has been an expansion of their functionality, which was largely known to be chemosensory related. However, more and more studies also indicated the non-chemosensory roles of insect CSPs. Therefore, it becomes growingly crucial to identify and characterize insect CSPs, which will assist in the comprehensive functional examination of CSPs. In this study, we identified 27 CSPs from *H. armigera*. We also performed in-depth characterization of 27 *H. armigera* CSPs using *H. armigera* genome and transcriptome data.

Our phylogenetic analysis showed that most *H. armigera*–specific CSPs are grouped in the same scaffold, indicating that they may be duplicated or evolved from the same ancestor gene. Comparing to other lepidopteran species, *H. armigera* are one of the most polyphagous lepidopteran herbivore species, this group of *H. armigera*–specific CSPs may contribute to their polyphagous feeding behaviors.

The expression profile revealed that the current functional domain CSPs are prevalent in the digestion tissues (foregut, midgut, and hindgut) and contact tissues (mouthparts and cuticle/epidermis), which was not previously known. CSPs are highly expressed in chemosensory tissues such as antennae and mouthparts to act as carriers to deliver the ligands to target receptors. Beside chemosensory organs, CSPs have been detected in many other tissues. The salivary gland in hematophagous mosquitoes contains small proteins from the D7-related (D7r) family [35]. These proteins can bind cysteinyl–leukotrienes and multiple biogenic amines, namely serotonin as well as norepinephrine. HarmCSP21/22 and 26 are extremely highly expressed in *H. armigera* salivary glands, which may play roles similar to those of D7-related proteins to bind the plant compounds.

Interestingly, we found that HarmCSPs are highly expressed in the major gustatory tissues including tarsi of both male and female adults, suggesting that these CSPs may play significant roles in the gustation system, for example, taste-molecule-binding proteins. Insect tarsi are directly exposed to and perceive plant compounds when they land on the plants through the chemoreceptors on the tarsi. It was reported that 14 gustatory trichoid chemosensilla were identified from the fifth tarsomere of female-adult *H. armigera* prothoracic legs, which exhibited electrophysiology responses to sugars or amino acids [36]. Our previous study showed that a taste receptor in the adult tarsi can respond to proline, a plant stress compound [34]. Similarly, 11 CSPs were identified from the cDNA library of female tarsi of the swallowtail butterfly, *Papilio xuthus* [37]. All these studies suggest the CSPs abundant in insect tarsi may act as carriers for plant chemicals for perception and detection. Further functional characterization of these CSPs can be performed in vivo by using RNA interference (RNAi) or CRISPR technologies.

We also developed a new method for soluble recombinant expression *H. armigera* CSPs. To do this, we came up with our own media for bacterial culture and in vitro protein expression. This medium was named as M16 medium, which is a combination of Terrific broth and Super broth [38]. M16 alone when used resulted increased bacterial growth in comparison to standard LB broth (Appendix A). In accordance to increased cell growth, CSP expression was also observed to increase but mostly all in insoluble (inclusion body) form (Appendix A). Thus, we decided to use glycerol as an additive in M16 media, for increasing the soluble expression of CSPs rather than in insoluble form. Previously, glycerol has been shown to be a very promising candidate to use as an additive media component in order to enhance the chances of soluble expression of recombinant proteins using bacterial cells [39]. For example, it has been shown that, when used as a media component, glycerol works as molecular chaperon and helps in obtaining a high yield of solubly expressed human phenylalanine hydroxylase using *E. coli* as the expression host [39]. Similarly, glycerol is also observed to be a carbon source for *E. coli* that has slow intake speed and, thus, helps in reducing the burden of acetate accumulation in cells by providing sufficient time to activate the carbon-stress-based acetate recycling mechanism [40,41,42]. In support to these reports, we obtained similar results showing beneficial effects of glycerol as a molecular chaperon that not only stimulates high cell growth but also upregulates the soluble expression of recombinant HarmCSP6 and 22 in *E. coli* cells.

In summary, we identified 27 *H. armigera* CSP genes and analyzed their expression maps in different insect tissues, stages, and sexes. We cloned HarmCSP6 and HarmCSP22 and developed a new approach to highly express them as soluble proteins in an *E. coli* system. These two recombinant HarmCSPs were further purified and characterized by using analytical techniques such as SE-FPLC, RP-HPLC, and MALDI-TOF. This newly developed protein expression method may be used broadly on other insect CSPs to obtain purer and more soluble proteins. A reverse chemical ecology approach will be applied on these two proteins to identify their ligands in future [23,43].

## Figures and Tables

**Figure 1 insects-13-00029-f001:**
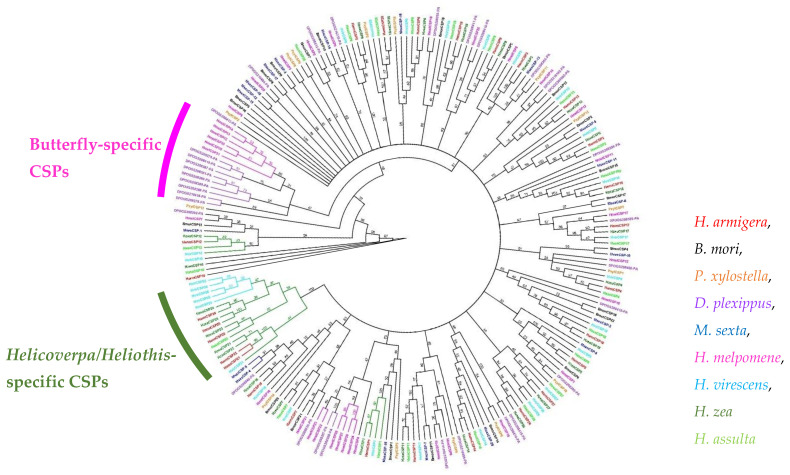
The maximum-likelihood phylogenetic analysis of selected lepidopteran CSPs from: *H. armigera*, *B. mori*, *P. xylostella*, *D. plexippus*, *Manduca sexta*, *H. melpomene*, *Heliothis virescens*, *H. zea,* and *H. assulta* (Appendix A). The amino acids of these CSPs are given in the supplementary data.

**Figure 2 insects-13-00029-f002:**
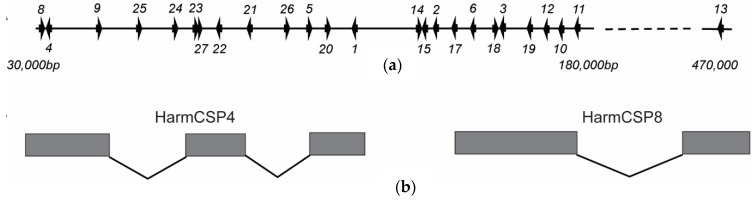
The schematic representation and structural analysis of *H. armigera* CSPs. (**a**) The location on scaffold 119 and (**b**) exon/intron structures of 25 *H. armigera* CSP genes. Introns are labeled using full lines while exons are representted using solid blocks. HarmCSP8 and 4 are used as examples for two groups of CSPs in exon/intron structures.

**Figure 3 insects-13-00029-f003:**
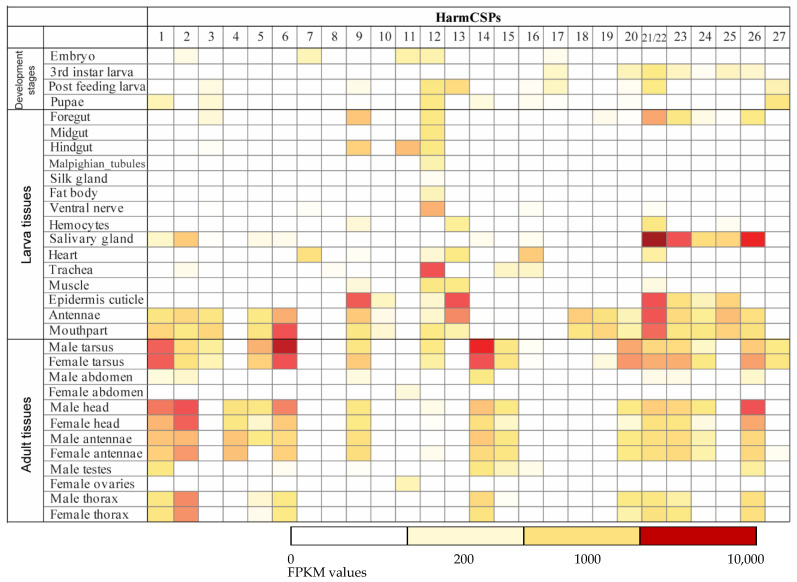
Expression profiles of *H. armigera* CSP genes. In silico expression of annotated HarmCSPs in studied tissues including: mouthparts, antennae, fat body, epidermis, malpighian tubules, foreguts, midguts, hindguts, hemocytes, trachea, hearts, silk glands, ventral nerve, salivary glands, and muscle of the 5th instar larvae; female heads, male heads, female antennae, male antennae, female thorax, male thorax, female tarsi, male tarsi, female testes, male ovaries, female abdomens, and male abdomens from day 0 to day 5 *H. armigera* adults; pupae, post-feeding stage larvae (whole body), 3rd instar larvae (whole body), and embryo. The color in the box indicates the level of FPKM value: Dark red, FPKM ≥ 10,000; yellow, FPKM = 200; and white, FPKM < 30.

**Figure 4 insects-13-00029-f004:**
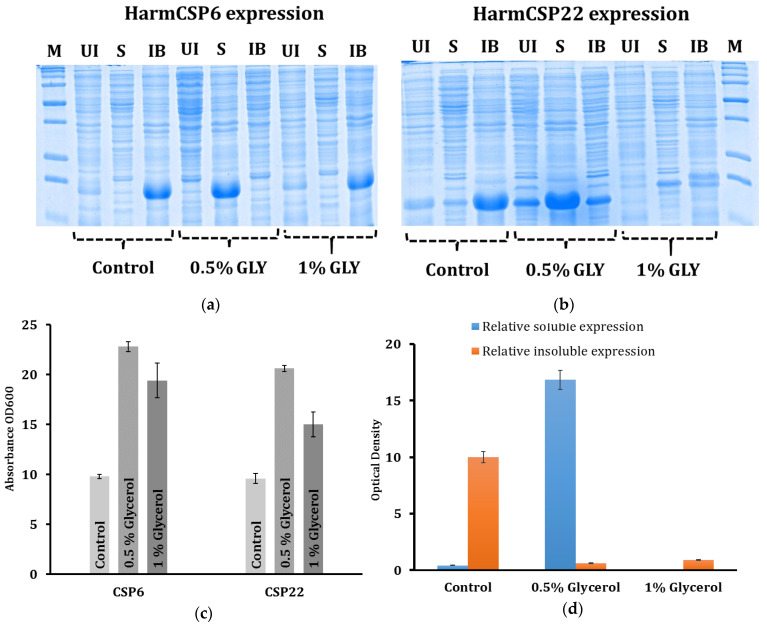
In vitro expression study of HarmCSP6 and HarmCSP22 using bacterial host. (**a**,**b**) SDS-PAGE gel images showing expression levels of HarmCSP6 and 22 in BL21-DE3 *E. coli* cells, respectively. UI—uninduced cells, IB—inclusion bodies, S—supernatant, control (M16 + 0% glycerol), 0.5% GLY (M16 + 0.5% glycerol), and 1% GLY (M16 + 1% glycerol). (**c**) Chart showing levels of absorbance at 600 nm of bacterial cell culture expressing HarmCSP6 and 22 in M16 media with 0%, 0.5%, or 1% glycerol. (**d**) Densiometric quantification of protein band showing relative soluble or insoluble expression of HarmCSP6 and 22 in M16 media with 0%, 0.5%, or 1% glycerol.

**Figure 5 insects-13-00029-f005:**
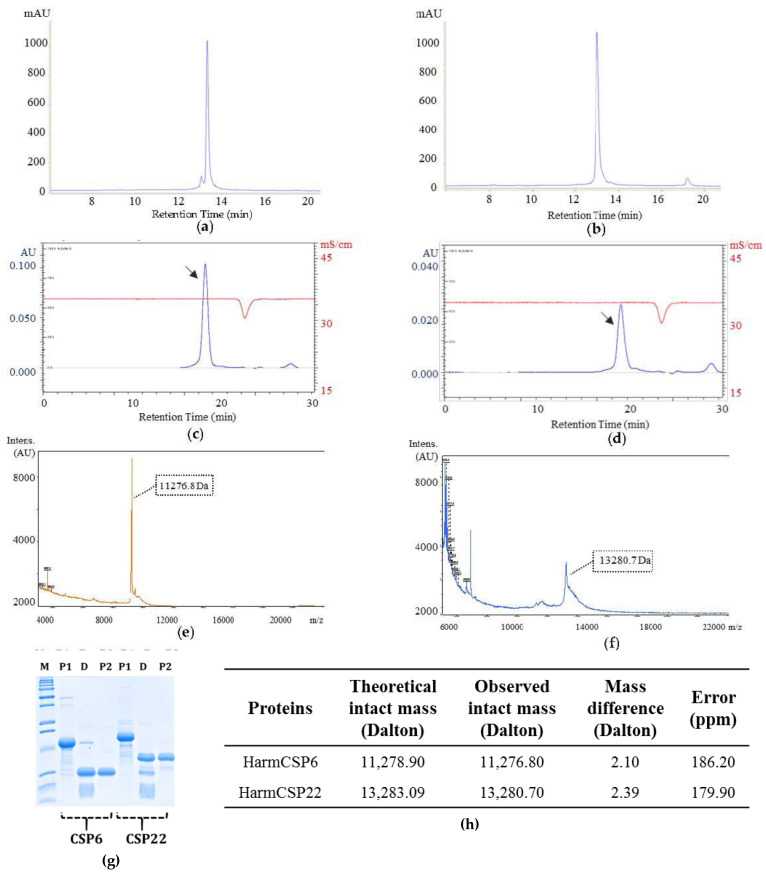
Purification and characterization of HarmCSP6 and 22. (**a**,**b**) RP-HPLC chromatogram of purified and digested HarmCSP6 and 22 showing >95% protein purity. (**c**,**d**) SE-FPLC chromatogram of purified and digested HarmCSP6 and 22 confirming the monomer nature (indicated by black arrow) of proteins; red line in both the chromatogram indicates conductivity. (**e**,**f**) MALDI-TOF intact mass profile of HarmCSP6 and 22. (**g**) SDS-PAGE gel image showing the Ni-NTA affinity chromatography of purified HarmCSP6 and 22. Lane 1—marker, Lane 2—first purification of HarmCSP6 (P1), Lane 3—enterokinase digestion of HarmCSP6 (D), Lane 4—second purification of HarmCSP6 (P2), Lane 5—first purification of HarmCSP22 (P1), Lane 6—enterokinase digestion of HarmCSP22 (D), Lane 7—second purification of HarmCSP22 (P2). (**h**) Table summarizing the MALDI-TOF intact mass profiles of HarmCSP6 and 22. For both HarmCSPs, the observed and theoretical intact mass is given in Daltons. Difference between these two masses is represented in the form of Error (ppm) to state the statistical significance of the data.

## Data Availability

The datasets used and/or analyzed during the current study are available from the corresponding author on reasonable request.

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
