# Peer review of "Chemosensory Proteins (CSPs) in the Cotton Bollworm Helicoverpa armigera"

_insects, 2021, doi:10.3390/insects13010029_

Round 1

Reviewer 1 Report

This research article is a result from a laboratory study on Chemosensory Proteins (CSPs) in the Cotton Bollworm Helicoverpa armigera. It was identified CSP genes from H. armigera, including ones. The study also analyzed CSP genes expression patterns through transcriptome data from different development stages and tissues of H. armigera. The authors found that H. armigera CSP genes are not only expressed in chemosensory but also in non-chemosensory tissues. The novelty here are the 27 CSP genes identified and a new method developed to express and purify recombinant CSPs. The study is well written, but it lacks a clear objective and pratical importance of the results. Overall, the study lacks a experimental data for an in-depth insight into the insect bahaviour, biology, physiology or ecology. In my opinion, the manuscript can be improved in clarity and conciseness if the authors are willing to consider the following suggestions.

Major comments:

  • The manuscript should be line-numbered in this stage to easy the review.
  • The study lacks a clear question nor shows what are the hypotheses are being tested.

Minor comments:

  • Simple summary – last phrase: “which” is in a different style from the rest.
  • Introduction – last paragraph: It should be clearly shown what are the questions and work hypotheses.
  • Introduction – last paragraph:“which may be the underlying mechanism why it is such a destructive polyphagous herbivore” How can the results help to underlying such mechanisms? It is not clear to me.
  • Double-check in the whole manuscript (including figures) the nomenclature of Genera and Species; most are not in Italic.
  • Material and methods, section 2.1.: instead of providing the name of a person who provided the insects, provide details on the source of the original insect colonies as well as the dynamics. For how long and what generation did you obtain them and if they were supplemented by wild colonies to prevent inbreeding or losing original traits. This is because insects kept over generation have the tendencies of changing their physiology in comparison to those in the wild.
  • Material and methods, section 2.2.: last paragraph is in different styles.
  • Results, page 11: I found this part a bit confusing as it mixes Materials and Methods and Results.

Author Response

We want to thank both reviewers’ insightful comments and suggestions, which make this manuscript much better. We also rewrote the manuscript to avoid the similarities to other published papers and reorganized the references. We followed both reviewers’ comments and revise the whole manuscript (marked in red colour) and our responses were below:

Major comments:

  • The manuscript should be line-numbered in this stage to easy the review.

Done!

  • The study lacks a clear question nor shows what are the hypotheses are being tested.

Done! We added one new paragraph in the introduction to explain why the newly developed protein expression method is important for the study on CSPs. Furthermore, in the last paragraph in introduction, we added the aims for this study.

Minor comments:

  • Simple summary – last phrase: “which” is in a different style from the rest.

Done!

  • Introduction – last paragraph: It should be clearly shown what are the questions and work hypotheses.

Done! We added one new paragraph in the introduction to explain why the new protein expression method is important for the study on CSPs. Furthermore, in the last paragraph, we added the aims for this study.

  • Introduction – last paragraph:“which may be the underlying mechanism why it is such a destructive polyphagous herbivore” How can the results help to underlying such mechanisms? It is not clear to me.

This sentence was deleted!

  • Double-check in the whole manuscript (including figures) the nomenclature of Genera and Species; most are not in Italic.

Done!

  • Material and methods, section 2.1.: instead of providing the name of a person who provided the insects, provide details on the source of the original insect colonies as well as the dynamics. For how long and what generation did you obtain them and if they were supplemented by wild colonies to prevent inbreeding or losing original traits. This is because insects kept over generation have the tendencies of changing their physiology in comparison to those in the wild.

We added the information in materials. Here the insects were used to extract RNA for CSP gene cloning. The insects for transcriptome data was provided the reference.

  • Material and methods, section 2.2.: last paragraph is in different styles.

Done

  • Results, page 11: I found this part a bit confusing as it mixes Materials and Methods and Results.

             Rewrite. We deleted the material and method parts from results.

Reviewer 2 Report

In this work, Agnihotri et al. identified 27 chemosensory proteins (CSPs) in Cotton Bollworm Helicoverpa armigera, and analyzed the expression of them using the RNA-seq data, and further developed a method to highly express recombinant CSPs in E. coli. The results seem interesting and provide new insights of CSPs in insect chemosensory system of H. armigera. However, I have severals concerns on the manuscript.  I would support it publish in the journal in case of these concerns were fully addressed.

  1. Since there were no line numbers in the MS, I suggest the authors add it in the revised MS. In the CSP phylogenesis part, Helicoverpa/Heliothis specific CSP group identified needs more lineage species, such as H. zea and H. assulta. Comparison of CSPs among these closely-related species would lead to more convinced evidence. Moreover, bootstrap value of the branches should supply  for a robust phylogenetic tree.
  2. Another major concern, the author claimed they developed a methos for large mounts of soluble recombinant CSP protein purification with M16 media and 0.5% glycerol, however, with no control of LB media culturing and only two CSP recombinant protein as example, it's not  rigorous to consider this approach as highly expression system.
  3. Writing. In. the results part, why did HarmCSP6 and HarmCSP22 were selected for protein expression? More details are needed for easy reading. In addition, so many descriptions of method contents in Page 9 to 10, which should move to 3.3. In-vitro Expression study of HarmCSP6 and HarmCSP22. This should be re-organized with full improvements. More numbers of CSPs may be replaced with More CSPs.
  4. Figure 3. A scale bar for color of the FPKM value would be better for comparison. 
  5. References was missing after the sentence" glycerol has been shown to be a very promising candidate to use as an additive..."

Author Response

We want to thank both reviewers’ insightful comments and suggestions, which make this manuscript much better. We also rewrote the manuscript to avoid the similarities to other published papers and reorganized the references. We followed both reviewers’ comments and revise the whole manuscript (marked in red colour) and our responses were below:

In this work, Agnihotri et al. identified 27 chemosensory proteins (CSPs) in Cotton Bollworm Helicoverpa armigera, and analyzed the expression of them using the RNA-seq data, and further developed a method to highly express recombinant CSPs in E. coli. The results seem interesting and provide new insights of CSPs in insect chemosensory system of H. armigera. However, I have severals concerns on the manuscript.  I would support it publish in the journal in case of these concerns were fully addressed.

  1. Since there were no line numbers in the MS, I suggest the authors add it in the revised MS.

Done!

  1. In the CSP phylogenesis part, Helicoverpa/Heliothis specific CSP group identified needs more lineage species, such as H. zea and H. assulta. Comparison of CSPs among these closely-related species would lead to more convinced evidence. Moreover, bootstrap value of the branches should supply  for a robust phylogenetic tree.

Done! We rebuit a new phylogenetic tree including H. zea and H. assulta CSPs. We also added the bootstrap values.

  1. Another major concern, the author claimed they developed a methos for large mounts of soluble recombinant CSP protein purification with M16 media and 0.5% glycerol, however, with no control of LB media culturing and only two CSP recombinant protein as example, it's not  rigorous to consider this approach as highly expression system.

Growth and expression of both CSPs were carried out in LB to compare it with the M16 expression study. The respective figures (Supplementary data Fig.S1 and Fig.S2) is attached below. Based on data it can be seen that, for CSPs, the recombinant expression was almost less than half in LB media, as compared to it in M16 media, and most of it was in an insoluble form. 

  1. Writing. In. the results part, why did HarmCSP6 and HarmCSP22 were selected for protein expression? More details are needed for easy reading.

The reasons that HarmCSP6 and HarmCSP22 were selected for expression were added in the results.

  1. In addition, so many descriptions of method contents in Page 9 to 10, which should move to 3.3. In-vitro Expression study of HarmCSP6 and HarmCSP22. This should be re-organized with full improvements.

Done!

  1. More numbers of CSPs may be replaced with More CSPs.

Done!

  1. Figure 3. A scale bar for color of the FPKM value would be better for comparison. 

A scale bar was added.

  1. References was missing after the sentence" glycerol has been shown to be a very promising candidate to use as an additive..."

The reference was added.

Round 2

Reviewer 1 Report

I assume that authors have made the changes proposed by reviewers and addressed all concerns raised in the first draft.

Sincerely,

Author Response

We thank the reviewers' comments and support, which made our manuscript much better. 

Reviewer 2 Report

The authors have done a good job, I think I have no reason to refuse it publish in the Journal. There is just one small concern, I suggest the bootstrap value at the branches keep the whole number.

Author Response

We thank the reviewers' comments and we prepared a new phylogenetic tree. The bootstrap values were revised to keep only whole numbers. Thanks.